# MULTI-AGENT REINFORCEMENT LEARNING FOR EFFICIENT VISION TRANSFORMER WITH DYNAMIC TOKEN SELECTION

## ABSTRACT

Vision Transformers (ViT) have revolutionized the field of computer vision by leveraging self-attention mechanisms to process images. However, the computational cost of ViT increases quadratically with the number of tokens. Dynamic token selection methods which aims to reduce computational cost by discard redundant tokens during inference, are primarily based on non-differentiable binary decisions methods and relaxations methods. However, Reinforcement Learning( (RL) based methods, which have astonishing decision-making ability, is considered to have high variance and high bias, not adopted for dynamic token selection task in previous work. Yet, RL-based methods have been successfully applied to many binary decision problems such as neural pruning, routing, path selection. In this paper, we propose *Reinforcement Learning for Dynamic Vision Transformer (RL4DViT)*, a novel framework for the dynamic token selection task in ViT using RL. By harnessing the powerfull decision-making capabilities of Multi-Agent Reinforcement Learning(MARL) algorithms, our method dynamically prunes redundant tokens based on input complexity, significantly reducing the computational cost while maintaining high accuracy. Extensive experiments on the ImageNet dataset indicate that our approach reduces the computational cost by up to 39%, with only a 0.17% decrease in accuracy. To the best of our knowledge, this is the first RL-based token selection method for efficient ViT. All our source code is publicly available at [link].

## 1 INTRODUCTION

*Vision Transformers (ViT)* have achieved state-of-the-art performance in many computer vision tasks while negating the need for convolution operations. However, they utilize a fixed token sequence independent of the input during both training and inference. This fail to take into account the fact that the complexity of a given computer vision task often varies depending on the input. For instance, distinguishing a flower from other categories is generally easier than differentiating between two similar species of flowers. Even within a single image, tokens containing detailed object features are far more informative than those from the background. Previous work Chefer et al. (2021) has shown that the final prediction of ViT is often based on a subset of the most informative tokens, suggesting that many tokens can be removed without significantly affecting performance. Motivated by this insight, we propose a framework that dynamically prunes redundant tokens based on input complexity, with minimal impact on task performance. By discarding redundant tokens in simpler cases and retaining more tokens for more complex ones, our approach significantly reduces the computational cost of ViT.

Several works have explored dynamic token selection in the ViT. For instance, DynamicViT Rao et al. (2021) introduces decision layers after each transformer block, trained with a loss function that considers both accuracy and computational cost. A-ViT Yin et al. (2021) employs a halting mechanism to discard less informative tokens without additional parameters. Moreover, A-ViT Wang et al. (2021) adds decision layers that determine not only which tokens to discard but also which attention heads and Transformer blocks to activate. Since dynamic inference in ViT is often based on binary decisions, which are non-differentiable, these works employ the Gumbel-Softmax

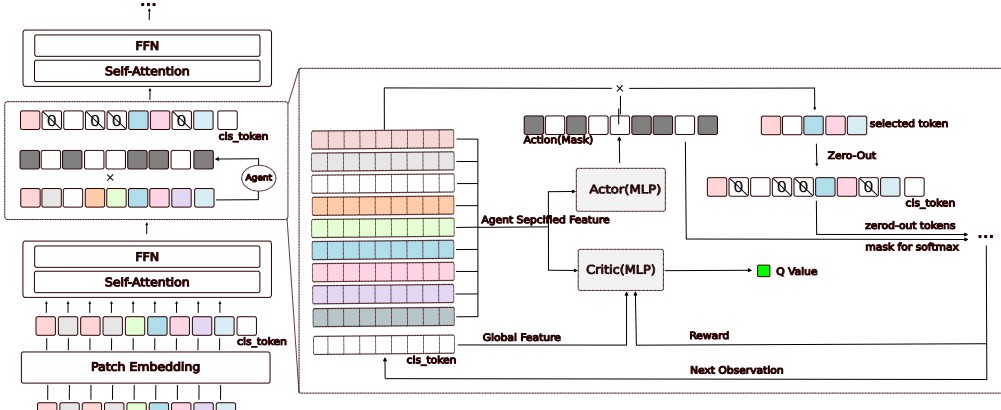

Figure 1: *Reinforcement Learning for Dynamic Vision Transformers(RL4DViT)*. This architecture incorporates *Multi-Agent Proximal Policy Optimization (MAPPO)* with ViT, where each agent corresponds to a token. The framework models the token selection process as a Markov Game, allowing agents to decide whether to retain or discard tokens based on their relevance to the input image. This dynamic pruning mechanism significantly reduces computational costs without a substantial loss in accuracy by eliminating redundant tokens while preserving critical information.

trick to enable end-to-end training. An alternative solution to this task involves using Reinforcement Learning (RL) to optimize the decision network through policy gradient methods.

Previous work on dynamic token selection in Vision Transformers has largely favored the Gumbel-Softmax trick, claiming that RL-based methods converge slowly due to high variance in the training signal, which scales with the dimensionality of the discrete variables. However, RL-based methods have been successfully applied to binary decision problems, including neural pruning Lin et al. (2017), behavior change Odena et al. (2017), dynamic routing Wang et al. (2018), and path selection Wu et al. (2018). While Gumbel-Softmax-based approaches may suffer from challenges such as regularization difficulties, stochasticity in training, and premature convergence, RL-based methods have demonstrated superiority in many cases. Therefore, we challenge the notion, propagated by previous studies, that variance and bias in RL algorithms hinder their application to dynamic token selection. This work seeks to explore the use of RL-based methods, such as Multi-Agent Proximal Policy Optimization (MAPPO), for dynamic token selection.

In this paper, we introduce *Reinforcement Learning for Dynamic ViT (RL4DViT)*, an RL-based dynamic token selection framework for the ViT. The overall structure of this framework is illustrated in Figure 1. RL4DViT consists of two parts: The ViT for inference and the MAPPO for token selection. When the input tokens in the ViT reach a specified block, we employ the MAPPO algorithm to determine whether a subset of these tokens should be discarded. The decision is then communicated back to the ViT. Based on this feedback, masked attention is applied as the tokens traverse the block, cutting off the connections between the tokens discarded by MAPPO and the remaining tokens. To the best of our knowledge, this is the first work to incorporate reinforcement learning for efficient Vision Transformer architectures with dynamic token selection.

Our main contributions are as follows:

- We propose RL4DViT,a novel reinforcement learning-based framework for dynamic token selection in Vision Transformers. By leveraging RL theroy, we model token selection task as a Markov Game, enabling efficient and adaptive pruning of redundant tokens based on input complexity.

- We conduct extensive experiments on the ImageNet dataset, demonstrating that RL4DViT significantly reduces computational costs by up to 39% while maintaining high accuracy, with only 0.17% drop in performance. Our results establish RL4DViT as a state-of-the-art solution for efficient ViT with dynamic token selection.

- We provide detailed visualizations of the token selection process, highlighting how RL4DViT discards redundant tokens early in the model's inference stages. This visualization demonstrates the efficiency of our approach in identifying and removing non-essential tokens.

The rest of this work is organized as follows: Section 2 discusses previous work on dynamic token selection task in ViT. Section 3 presents the pipeline of applying MAPPO to the dynamic token selection task. Section 4 reports the experimental results on ImageNet 1k. Section 5 concludes this paper.

## 2 RELATED WORK

### 2.1 DYNAMIC TOKEN SELECTION IN VISION TRANSFORMER

Inference speed is crucial for deploying deep models on edge devices. Numerous techniques can accelerate the inference speed of deep models, including quantization Osmanović (2021); Jacob et al. (2018), pruning Frankle & Carbin (2018), and knowledge distillation Gou et al. (2021); Pham et al. (2022). Several works also focus on accelerating the inference speed of transformer models. For instance, TinyBERT Jiao et al. (2019) introduces a distillation method to speed up transformer inference, while Swin-Transformer Liu et al. (2021) reduces the quadratic space and time complexity to linear by replacing the fully connected structure with a star-shaped topology. However, most of these works focus on natural language processing (NLP) tasks.

Our method, on the other hand, dynamically prunes less important tokens by exploiting the sparsity of informative image patches in Vision Transformers. The key to implementing dynamic inference in Vision Transformers is to prune tokens with less information as much as possible while maintaining model performance. Serveral techniques can be employed for this task, such as the Gumbel-Softmax Jang et al. (2016) trick, halting mechanisms Yin et al. (2021), and RL-based methods. For example, DynamicViT addresses this issue by adding decision layers after each transformer block, training these layers using a loss function designed to balance both accuracy and computational cost Rao et al. (2021). Similarly, A-ViT Wang et al. (2021) implements a halting mechanism to discard less informative tokens without adding extra parameters, modifying the Vision Transformer architecture in the process. Moreover, AdaViT introduces decision layers to not only determine which tokens to discard but also which attention heads and transformer blocks to activate. Unlike Gamble-Softmax trick and halting mechanisms, RL-based methods has not yet been applied to this task, even though it has strong decision-making capabilities.

## 3 THE PROPOSED FRAMEWORK

We propose RL4DViT, a MARL-based adaptive computation framework designed to reduce the computational cost of ViT with minimal impact on performance. Given an input image, RL4DViT learns to adaptively derive policies for selecting which tokens to discard or retain in the Vision Transformer backbone, conditioned on the input image. An overview of our method is illustrated in Figure 1. In this section, we present how to apply MARL techniques, such as MAPPO, to the dynamic token selection task in Section 3.1. Then, we elaborate on reward engineering and attention masks in Sections 3.2 and 3.3.

### 3.1 DYNAMIC TOKEN SELECTION WITH MAPPO

We aim to integrate dynamic token selection with MAPPO by mapping each token in the ViT to an agent in the MAPPO framework. By representing each token as an agent, we can make decisions on whether to discard or retain the token.

After establishing the mapping between tokens and agents, we proceed to model the dynamic token selection process in ViT as a Markov Game within the MAPPO framework. From a multi-agent reinforcement learning perspective, we consider the computation process of all tokens $T_i (i = 0, 1, ...t)$ from the first transformer block to the last transformer block. Consequently, we model the dynamic token selection process as a Markov Game, defined as follows:

States (S) : We denote a state by $s_i$. We define the state of current environment as the token vector $T_i$ itself, which means one agent can only observe one single token and the computation process of this single token as its observation as State $s_i$.

Joint Action (A) : We denote joint action as $A = (a_1, a_2, ..., a_i)$, We define two actions, namely $a_i = 0$ and $a_i = 1$. The former means to discard token $T_i$, and the latter means to keep token $T_i$.

Transitions (T) : Given a current state $s_i$ and joint actions A to take, the probability that we would observe a specific state $s_i'$ is $P(s'|s, a_1, a_2, ..., a_i)$.

Rewards (R) : We denote a reward for agent $i$ as $R_i(s, a_1, a_2, ..., a_i)$, the reward associated with the transition from the state $s_i$ to state $s_i'$, after joint action action $A$ is taken, a scalar reward was given by the reward function, which will be illustrate in section 3.2. With the reward defined by reward enginerring, the goal for MAPPO Algorithm is to maximize the accumalated rewards:

$$\mathbb{E}[\sum_{t=0}^{\infty} \gamma^l R_i(s_l, a_1^l, a_2^l, ..., a_i^l)]$$

where $\gamma$ is the discount factor and $l$ is the time step, whcich is the $l_{th}$ decision block in ViT.

The detailed training procedure of MAPPO for dynamic token selection is presented in Algorithm 1, which we go through as follows.

We begin by initializing the actor network $\pi_{\theta_t}(a|s)$ with random weights $\theta$ and the critic network $V_\phi(s, a)$ with random weights $\phi$. Next, we configure a series of hyperparameters for the RL algorithm, including the learning rates for both the actor and critic networks, the Adam optimizer's $\epsilon$ parameter, and the maximum number of training steps. Subsequently, we initialize ViT with pre-trained weights, keeping these weights fixed while training our MARL algorithms.

With all the initial preparations completed, we can begin sampling trajectory data for the training process of our MARL algorithm. Using the symbols defined earlier, a trajectory can be expressed as follows:

$$s^0, A^0, r^0, s^1, A^1, r^1, ..., s^l, A^l, r^l, ...$$

where $s^l$ is the token vectors before transformer block $l(l = 0, 1, ..., L)$, $A^l$ means the union action of made by all agents before transformer block $l$.

To sample trajectory data, the inference process of vision transformer need to be started first. To achieve this, a batch of image which batch size equas to $B$ as input $I$ were send to ViT-base, the input $I$ will be split into 196 tokens and one $cls$-token was appended so that $T = 197$. For each token $t$ in input $I_{B \times T}$, before those token flow through transformer block $l$, we token the token vetor $t$ (in this case, the dimension of $t$ is 768) as $s_l^{bt}$, a decision of to determine whether to keep or discard token $t$ was achived by following $a_l^{bt} \sim p_l^{bt} = \pi_\theta(s_l^{bt})$. The decision is executed by masked attention mechanism when the tokens flow through vision transformer block $l$ and the output of block $l$ is taken as $s_{l+1}^{bt}$, in the same way, we can derive $a_{l+1}^{bt}$ from $s_{l+1}^{bt}$. Thus, we can get the decision trajectory for each token by interact actor network $\pi_\theta$ with vision transformer.

After sampling process, we use minibatch data sampled from data buffer $D$ to train our actor network $\pi_\theta$ and critic network $V_\phi$.

The actor network is trained to maximize the loss function:

$$L(\theta) = \frac{1}{mn} \sum_{i=1}^{m} \sum_{t=1}^{n} \min \left( r_{\theta,i}^{(t)} \hat{A}_i^{(t)}, \text{clip} \left( r_{\theta,i}^{(t)}, 1 - \epsilon, 1 + \epsilon \right) \hat{A}_i^{(t)} \right) + c \cdot \frac{1}{mn} \sum_{i=1}^{m} \sum_{t=1}^{n} S \left[ \pi_\theta \left( o_i^{(t)} \right) \right]$$

where $m$ is size of minibatch, $t$ is number of agents, $\hat{A}_i^b$ is computed using the GAE method, $S$ is the policy entropy, and $c$ is the entropy coefficient hyperparameter, $r_{\theta,i}^{(t)} = \frac{\pi_\theta \left( a_i^{(t)} | o_i^{(t)} \right)}{\pi_{\theta_{\text{old}}} \left( a_i^{(t)} | o_i^{(t)} \right)}$.

The citic network is trained to minimize the loss function:

$$L(\phi) = \frac{1}{mn} \sum_{i=1}^{m} \sum_{t=1}^{n} \max \left[ \left( V_\phi \left( s_i^{(t)} \right) - \hat{R}_i \right)^2, \left( \text{clip} \left( V_\phi \left( s_i^{(t)} \right), V_{\phi_{\text{old}}} \left( s_i^{(t)} \right) \pm \varepsilon \right) - \hat{R}_i \right)^2 \right]$$

After the training process of the reinforcement learning algorithm is completed, we fine-tune the weights of ViT while keeping the MAPPO weights fixed. This approach is intended to mitigate the influence of distribution shifts in the ViT caused by token selection.

## 3.2 REWARD ENGINEERING

We developed two reward functions to assist the agents in optimizing the trade-off between computational cost and accuracy:

$$reward_{1i} = \begin{cases} 0, \text{ if agent } i \text{ is died} \\ -1, \text{ if agent } i \text{ is alive} \end{cases} \quad reward_{2i} = \begin{cases} 0, \text{ if classify wrongly} \\ R/n, \text{ if classify correctly} \end{cases}$$

where $R$ is a variable designed to encourage the agents to discard tokens that contain redundant or noisy information, and $n$ represents the total number of active tokens, motivating the agents to discard as many tokens as possible.

the total reward is:

$$R_i = \alpha * reard_{1i} + \beta * reward_{2i}$$

where $\alpha$ and $\beta$ are parameters used to balance the trade-off between computational cost and accuracy. Additionally, although these agents are homogeneous, their relationships are both competitive and cooperative. Therefore, we do not use a shared reward for the agents.

---

**Algorithm 1** RL4DViT

---

**Input:** $input$
**Output:** $output$
1: initialize $\theta$, the parameter for policy $\pi$. intialize $\phi$, the parameter for critic $V$
2: set learning rate $\alpha$
3: initialize vision transformer $m$ with pretarined weights.
4: **while** $step \leq step_{max}$ **do**
5:     set data buffer $D = \{\}$
6:     **for** b = 0 to B **do**
7:         $\tau = []$ empty list
8:         **for** l = 0 to L **do**
9:             observe $s_l^b$
10:             **for** agent $t$ **do**
11:                 $p_l^{bt} = \pi_{\theta_t}(s_l^{bt})$
12:                 $a_l^{bt} \sim p_l^{bt}$
13:                 $A_l^b = concat : a_l^{bt}$
14:             **end for**
15:             execute action $A_l^b$, observe $r_l^b, s_{l+1}^b$
16:             $\tau + = [s_l^b, A_l^b, r_l^b, s_{l+1}^b]$
17:         **end for**
18:         compute advantage estimate $\hat{A}$ via GAE on $\tau$
19:         $D = D \cup \tau$
20:     **end for**
21:     Shuffle and split data buffer $D$ into minibatches of size $batch\_size$
22:     **for** each minibatch $m$ **do**
23:         compute $L(\theta)$ on minibatch $m$
24:         compute $L(\phi)$ on minibatch $m$
25:         Adam update $\theta$ with $L(\theta)$
26:         Adam update $\phi$ with $L(\phi)$
27:         learning ratet $\alpha$ decay
28:     **end for**
29: **end while**

---

### 3.3 TOKEN SELECTION WITH ATTENTION MASK

To execute action $A$ in the Vision Transformer, we need to mask out the tokens that have been discarded. We can simply prune the tokens that the MARL algorithm has decided to discard. However, the action $A$ is usually unstructured, and the actions for different samples are not the same. Therefore, for samples within a batch, simply pruning tokens where $A_t = 0$ would result in a non-uniform number of tokens, making it difficult to parallelize the computation. To maintain a consistent number of tokens across samples within a batch, we opt to zero out the values of the discarded tokens, which is akin to padding operations in NLP tasks. However, merely zeroing out those discarded tokens is insufficient, as these zeroed tokens can still affect other tokens through the softmax function in the calculation of the self-attention matrix Vaswani et al. (2017):

$$Attn = Softmax(\frac{QK^T}{\sqrt{d_k}})$$

One way to address this problem is to set the corresponding elements in the attention matrix to a negative value, such as -1000, after calculating the scores and before applying softmax. This approach ensures that the corresponding values after passing through softmax are almost 0 Yin et al. (2021). Alternatively, after applying softmax, we can set the corresponding elements in the attention matrix to 0 Rao et al. (2021). By doing so, we can effectively cuts the connection between the discarded tokens and the other tokens.

The masking strategy described above keeps the computational cost of our training iterations similar to that of the original Vision Transformer's training cost. During inference, we simply prune the discarded tokens from computation to measure the actual computational efficiency gained by our reinforcement learning algorithms.

## 4 EXPERIMENT

In this section, we will demonstrate the superiority of reinforcement learning-based dynamic token selection through extensive experiments. In all of our experiments, we fix the number of transformer blocks that adopt token selection to $L = 3$. Furthermore, inspired by previous work, the token selection is performed hierarchically in three stages: at block $3, 6$, and $9$. We train the MARL agents with ViT weights fixed for one epoch and fine-tune the ViT with MARL agents' weights fixed for another epoch. All the experiments are carried out on a single NVIDIA RTX 3090 GPU. More implement details can be found in the supplementary material.

RL4DViT can be trained very fast. While previous work had to train the entire network from scratch or train the selection network for many epochs, RL4DViT can achieve comparable performance within one epoch. Another advantage of RL4DViT is that the total number of parameters for token selection is much smaller. In DynamicViT, it was necessary to introduce MLP into three decision blocks. In A-ViT, it was necessary to implement an h-gate in each transformer block, while in RL4DViT, the network for token selection consists solely of the Actor (which is one MLP) with a total parameter count of 0.1M, reducing the number of parameters utilized for decision-making by a factor of 9 compared to DynamicViT.

### 4.1 MAIN RESULT

We summarize the main results on ImageNet 1k Deng et al. (2009) in Table 1. We report the top-1 accuracy under different model complexity(GFlops) . We demonstrate that RL4DViT can reduce the computational costs by 39%, with the influence on performance down to -0.5%. Furthermore, after eliminating the distribution shift by fine-tuning the parameters of ViT in RL4DViT, the influence on performance can be reduced down to -0.17%. To put it more intuitively, we can cover half of a picture during inference while the influence on performance is negligible.

### 4.2 FINE-TUNE

Different from previous work like DynamicViT or A-ViT, where the MLP or h-gate for token selection is trained alongside the Vision Transformer, RL4DViT trains the agents separately from the

Table 1: Comparisons with State of the art on ImageNet. We compare our RL4DViT models with state-of-the-art dynamic token selection models with comparable GFlops and number of parameters. We use Vit and RL4DViT as base model, we aslo include the results of DynamicVit and A-ViT as references. It's worth mentioning that by discarding a small portion of tokens, a higher accuracy than the original Deit-B can be achieved, implying that there may be noisy token present rather than just redundant token.

| Model | Parameters(M) | GFLOPs | Top-1 Acc(%) |
|---|---|---|---|
| Deit-T | 5 | 1.2 | 71.3 |
| Deit-S | 22.1 | 4.6 | 79.8 |
| Deit-B | 86.6 | 17.5 | 81.8360 |
| DynamicVit-T | 5.9 | 1.2 | 71.3 |
| DynamicVit-S | 23 | 3.4 | 78.3 |
| DynamicVit-B | 87.5 | 11.2 | 81.3 |
| A-Vit-T | 5 | 0.8 | 71.0 |
| A-Vit-T + distl. | 5 | 0.8 | 72.4 |
| MAPPO-Deit-B | 86.8 | 10.0 | 81.1360 |
| MAPPO-Deit-B | 86.8 | 10.8 | 81.3080 |
| MAPPO-Deit-B | 86.8 | 11.6 | 81.3860 |
| MAPPO-Deit-B | 86.8 | 11.8 | 81.6180 |
| MAPPO-Deit-B | 86.8 | 13.1 | 81.6380 |
| MAPPO-Deit-B | 86.8 | 14.7 | 81.6700 |
| MAPPO-Deit-B | 86.8 | 16.1 | 81.7680 |
| MAPPO-Deit-B | 86.8 | 16.6 | 81.8380 |
| MAPPO-Deit-B | 86.8 | 17.2 | 81.8760 |

training process of the Vision Transformer. The training data for the reinforcement learning algorithm is collected by executing token selection in the Vision Transformer with pretrained weights.

Table 2: The impact of fine-tuning on MAPPO-Deit-B after dynamic token selection. We fine tuned two RL4DViT models under different complexity. The results show that fine-tuning improves the Top-1 accuracy while maintaining lower computational costs (GFLOPs), demonstrating how fine-tuning helps mitigate the distribution shift caused by token pruning.

| Model | Parameters(M) | GFLOPs | Top-1 Acc(%) |
|---|---|---|---|
| Deit-B | 86.6 | 17.5 | 81.8360(-0.0) |
| DynamicVit-B | 87.5 | 11.2 | 81.3(-0.54) |
| MAPPO-Deit-B | 86.8 | 10.8 | 81.3080(-0.53) |
| MAPPO-Deit-B | 86.8 | 11.6 | 81.3860(-0.44) |
| MAPPO-Deit-B-FT | 86.8 | 10.8 | 81.5700(-0.27) |
| MAPPO-Deit-B-FT | 86.8 | 11.6 | 81.6620(-0.17) |

Training the RL agents alone in RL4DViT can make the training process very efficient. However, without updating the weights of the ViT, the input to the ViT after token selection undergoes a notable distribution shift, and the unupdated weights of the ViT can impair the performance of RL4DViT. Thus, after the training of the RL algorithm, we fine-tune the ViT to mitigate the adverse impact of the distribution shift.

## 4.3 VISULIZATIONS

To further investigate the token selection policy performed by RL4DViT, we visualize the token selection process in Figure 2. We show the original input image and the token selection results after the three token selection blocks, where the color of the patch (area) corresponds to the discarded tokens that were replaced with white.

We found that the selection policy performed by RL4DViT is much different from previous work. In contrast to the tokens discarded hierarchically in DynamicVit or A-Vit, which means tokens were

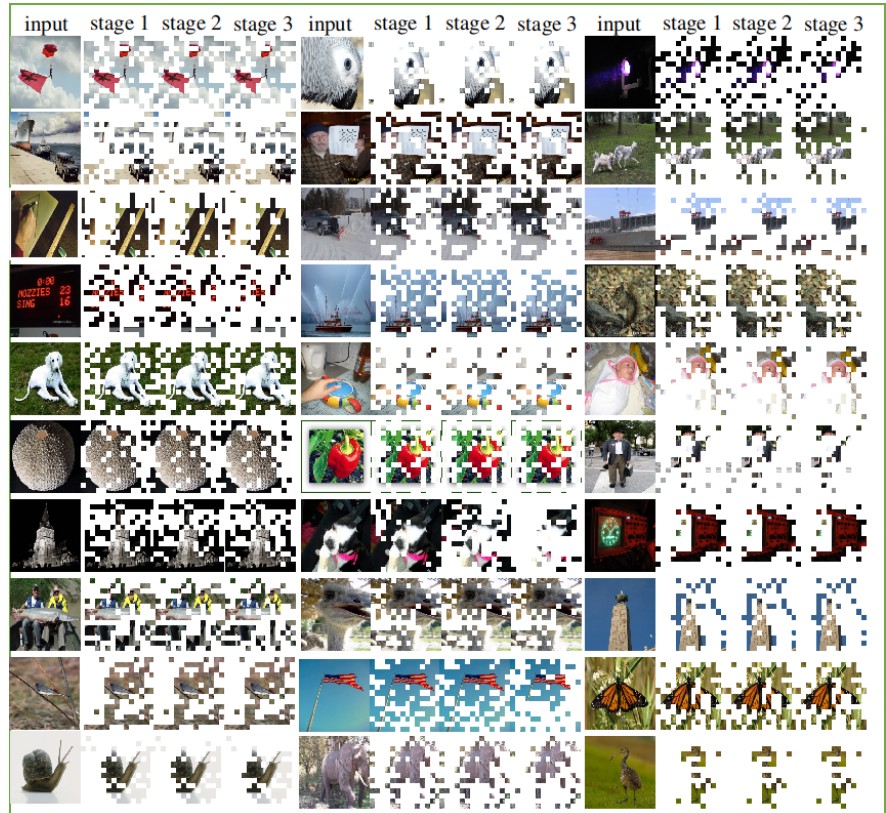

Figure 2: The original input image is displayed alongside the selection results from transformer blocks [3, 6, 9], with discarded tokens highlighted in white. The token selection process in RL4DViT contrasts with previous methods like DynamicVit and A-Vit. Unlike the step-by-step approach of earlier models, RL4DViT discards redundant tokens in bulk during the first selection stage, demonstrating its ability to swiftly identify and eliminate non-essential information. This method not only prioritizes background details near the subject but also discards low-information tokens within the subject itself, revealing the underlying patterns of redundancy in the data.

discarded step by step, RL4DViT discards as many tokens as possible in the first token selection block; the token selection policy remains almost static in the subsequent token selection blocks. In fact, such an extreme strategy aligns with intuition: if we can identify a token as redundant, we do not want to discard it step by step; on the contrary, we should discard it as soon as possible to save more computational cost. This extreme selection policy discovered by RL itself, compared to other step-by-step selection policies, means our algorithm can actually recognize a redundant token and uncover the underlying patterns of redundancy.

Besides the sample-wise visualization shown above, we are also interested in the underlying pattern of redundant tokens. In DynamicVit, tokens outside the outline of the subject are recognized as redundant, allowing DynamicVit to focus on the main object in the image. RL4DViT employs a similar token selection strategy with a slight difference: instead of just keeping tokens inside the outline of the subject like DynamicVit, RL4DViT also tends to keep tokens near the outline of the subject, which usually contain parts of the background information.

## 4.4 THE EFFECT OF REWARD FUNCTION PARAMETER TO TOKEN SELECTION

In the reward engineering, we use hyperparameters such as $\alpha$ and $\beta$ to balance MAPPO's trade-off between accuracy and computational cost. A larger $\alpha$ encourages the algorithm to select fewer tokens for better computational efficiency, while a larger beta encourages the algorithm to select

more tokens for better prediction accuracy. We explored the impact of different ratios of $\alpha/\beta$ on the token selection strategy in Figure 3.

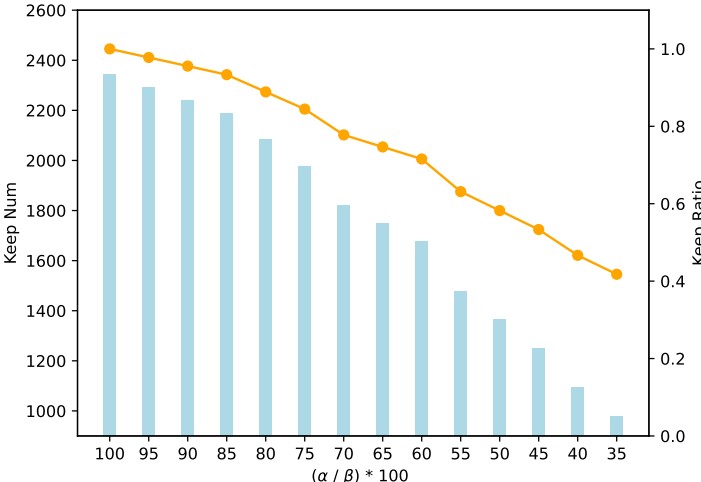

Figure 3: During inference in ViT, the total tokens needs to calculate besides cls token is 196*12 equals to 2352, through multiple evaluations, we calculated the total number of kept tokens under different ratios, and in general, as the ratio of $\alpha/\beta$ decreased, the algorithm tended to select fewer tokens.

## 4.5 COMPARE TO RANDOM POLICY

We show the comparasion between random token selection with different blocks in Figure 4. The token was randomly discard with ratio $[\rho, \rho^2, \rho^3]$. With same token keep ratio, the further forward the block is, the less accurate it is. We then compare the random token selection policy with the un-finetuned token selection policy under same model complexity(GFlops).

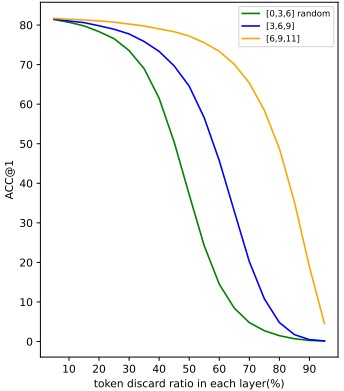
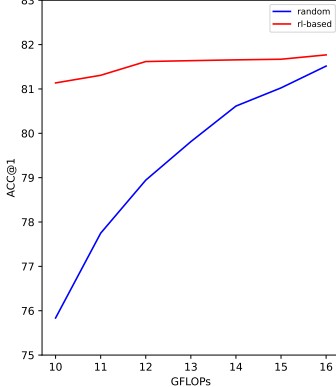

Figure 4: Comparison of Random vs. RL-Based Token Selection The graph shows the effect of token discard ratio at various layers of the Vision Transformer. It demonstrates that random token selection at different layers negatively impacts accuracy, while the RL-based token selection method consistently achieves higher accuracy across various model complexities (GFlops).

## 5 CONCLUSION

In this paper, we proposed the dynamic token selection framework RL4DViT, which significantly enhances the efficiency of ViT in computer vision tasks. By modeling the token selection task as a Markov game, we utilized Multi-Agent Reinforcement Learning (MARL) to dynamically adapt to input complexity. This method not only effectively reduces computational overhead by up to 39% but also results in only a 0.17% minor decrease in accuracy, demonstrating its feasibility and effectiveness in practical applications. Our research indicates that reinforcement learning-based strategies offer greater flexibility and efficiency in dynamic token selection, providing a new direction for future research in the field of Vision Transformer.

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

# A    IMPLEMENT DETAILS

## A.1    PARAMETER SHARING

Agents in MAPPO are homogeneous, meaning they possess identical observation and action spaces. We utilize parameter sharing to reduce the number of parameters in our RL4DViT framework. Additionally, previous works have demonstrated that this approach improves learning efficiency Christianos et al. (2021) Terry et al. (2020). In our implementation of MAPPO for dynamic Vision Transformer, the agents share both the policy and value function parameters.

## A.2    FEATURE CONSTRUTION

As mentioned earlier, MAPPO operates within the centralized training with decentralized execution (CTDE) framework, enabling individual PPO agents to communicate through a global value function. The input to the value function typically consists of the global state information $s_i^{(t)}$. This global state transforms a partially observable Markov decision process (POMDP) into a Markov decision process (MDP). Proper design of the global state information $s_i^{(t)}$ is crucial.

One approach is to concatenate the local observation information of all agents as the input for the critic. While this can be effective in most cases, in our scenario, where the number of agents is significant (in our case, $t = 196$), this may lead to an excessively high input dimension for the value function compared to the policy function, complicating the learning of value functions, as noted in Lowe et al. (2017).

An alternative is to use an Environment Provided global state (EP), which contains general global information about the environment state. However, the EP state typically encompasses information common to all agents and may overlook critical local agent-specific details Foerster et al. (2018). A

third approach is to construct an Agent-Specific Global State (AS), which generates a global state for agent $t$ by concatenating the EP state with the local observation of agent $t$, as discussed in the original MAPPO paper Yu et al. (2022).

In our case, we construct the Agent-Specific Global State by utilizing the class token in the Vision Transformer as the Environment Provided global state (EP). By concatenating $token_{cls}$ and $token_t$, we create the global state for agent $t$. Thus, the Agent-Specific Global State for agent $t$ is given by:

$$s_i^{(t)} = \text{concat}(token_t, token_{cls}),$$

and the dimension of $s_i^{(t)}$ is $768 \times 2$, which equals 1536.

## A.3   GAE

We adopt Generalized Advantage Estimation (GAE) Schulman et al. (2015) with advantage normalization and value clipping in the implementation of MAPPO.

## A.4   AGENT POSITION POLICY

The agents in MAPPO are homogeneous, and the number of agents (in our case, $n = 196$) is significantly higher than the usual number of agents in typical scenarios (which is 3 to 12). To help the agents understand their location in the environment, we add a one-hot agent ID to the observation for agents in MAPPO. This means the input to the critic and actor in MAPPO is the Agent-Specific Global State concatenated with the one-hot agent ID.

