# OpenReview forum: "Multi-Agent Reinforcement Learning for Efficient Vision Transformer with Dynamic Token Selection"
_ICLR.cc/2025/Conference — ICLR 2025 Conference Withdrawn Submission_

### Official Review · Reviewer_nnZF · 2024-10-21

**Soundness:** 3
**Presentation:** 3
**Contribution:** 3
**Rating:** 5
**Confidence:** 4

**Summary:**

The paper presents "RL4DViT", a novel framework for the dynamic token pruning task in ViTs based on multi-agent reinforcement learning methods. RL4DViT takes each image token as an agent and decides whether to retain or discard itself based on its vector. Within sequential ViT blocks, RL4DViT formulates a Markov Game, to maximize the reward (higher accuracy & lower computational cost). Extensive experiments validate that RL4DViT can reduce 39% computational cost with only a 0.17% top-1 accuracy decrease on ImageNet-1K, validating the effectiveness of the proposed method.

**Strengths:**

1. Modeling the dynamic token pruning task as a Markov Game is quite novel and reasonable.
2. Utilizing MAPPO to solve it makes sense.
3. The presentation of the algorithm is clear.
4. Compared with DynamicViT and A-ViT, the proposed method performs slightly better.

**Weaknesses:**

1. To address the proposed token selection by using RL, a straightforward approach is to optimize it with a single-agent reinforcement learning method. The input features can be regarded as the input state and the decision of whether to retain or discard each token can be regarded as the action. However, the authors didn’t explore single-agent RL methods or discuss the differences. It would be better for the authors to explain the reason for not exploring single-agent RL methods or analyzing the differences between single-agent and multi-agent RL methods.
2. Except for classification results on ImageNet-1K on ViT-B, there are no other datasets (CIFAR-10/100) or vision models (ViT-T/ViT-L/Swin). It would be better for the authors to validate RL4DViT on more datasets or vision models or explain the reasons for choosing only one experimental setting.
3. There are several outstanding token pruning methods that the authors didn’t compare or mention [1][2]. I think the authors should discuss the differences or the advantages of RL4DViT with more baseline methods.
4. Should the rewards of discarded tokens be the same? I think the tokens discarded at earlier stages should have higher rewards, which can cause more computation cost reduction. This is just my suggestion that may be helpful for further improvement and will not affect my ratings.

[1] Kong, Zhenglun, et al. "Spvit: Enabling faster vision transformers via latency-aware soft token pruning." European conference on computer vision. Cham: Springer Nature Switzerland, 2022.

[2] Bolya, Daniel, et al. "Token Merging: Your ViT But Faster." The Eleventh International Conference on Learning Representations.

**Questions:**

Please refer to weaknesses. Why should the authors use MARL, instead of single-agent RL? This is my main concern, and I will raise my ratings if the authors' response addresses it well.

---

> ### Author Response · Authors · 2024-11-13
>
> Dear Reviewer,
> I greatly appreciate your time and the suggestions you provided. Although the rating '5' was negative, it is still a tremendous encouragement to me.
>
> In response to your question about why I chose multi-agent RL over single-agent RL, here is my explanation: For an image with T  tokens, after a single agent makes a decision a , the state transition equation is  p(s', r | s, a) . However, in the forward inference process of ViT, the state transition equation is actually  p(s', r | s, A) , where  A  is the joint action of all agents. Simply put, during ViT’s forward inference, the state of  block_{i+1}  is determined by all tokens in block_i , not by any single token. In fact, I initially explored using single-agent RL for this task at the beginning of this work. However, the experimental results could not surpass the then state-of-the-art, DynamicViT. Based on these results, I reconsidered the theoretical limitations of applying single-agent RL to this task and switched to a multi-agent RL algorithm. To keep the paper focused on the multi-agent reinforcement learning algorithm, I removed the theoretical section and experimental results related to PPO from the draft.
>
> In response to your question regarding why Deit-B is the only backbone used, here is my explanation: The primary backbones used for comparison in DynamicViT and A-ViT are Deit-B, Deit-T, and few other variants. To ensure a fair comparison on the same dimension, I chose Deit-B, which was used in both of these papers. I acknowledge that the lack of comparison across multiple backbones limits the ability to demonstrate the algorithm's generality and versatility. I will carefully revise the paper based on your suggestions.
>
> In response to your concern about the baseline, I did indeed overlook some work, and I will address this in future revisions.
>
> I want to especially thank you for your suggestions regarding reward design. I am very happy to know that someone shares the same ideas as I do. In fact, during my exploration, I did try a reward aligned with your idea to encourage the algorithm to discard redundant tokens early. However, this reward, which was designed from the perspective of a single agent, led to slow convergence and oscillations in MAPPO. As an alternative, I designed a global reward, R/n , where R  is a constant,  n  is the number of retained tokens, and this design also encourages the algorithm to discard tokens early. In fact, our visualization results clearly demonstrate this: our algorithm prunes the majority of tokens it identifies as redundant in the first decision layer. This pruning strategy is markedly different from the gradual pruning approach used in DynamicViT and A-ViT, and it aligns more closely with our intuition—that if the algorithm can identify redundant tokens, it should remove them as early as possible.
>
> I have decided to withdraw my submission to make further revisions.Once again, I sincerely thank your kindness and your advice, and I wish you good health.

---

### Official Review · Reviewer_hBRZ · 2024-11-01

**Soundness:** 2
**Presentation:** 3
**Contribution:** 1
**Rating:** 3
**Confidence:** 5

**Summary:**

This paper proposes to utilize a multi-agent reinforcement learning (MARL) approach for the token selection process in token pruning for efficient ViTs. Specifically, a multi-agent proximal policy optimization (MAPPO) method is adapted to the token selections. The proposed method is validated on one ViT backbone and compared with existing token pruning methods.

**Strengths:**

1. __Interesting idea__: The idea of incorporating MARL into the token selection process is interesting and novel. As far as I know, this is the first work leveraging MARL for token reduction.

2. __Well written and organized__: This paper is well-written and organized. It's easy to read and follow.

3. __Clear explanations__: The introduction and explanations to the MARL method are clear.

**Weaknesses:**

1. __Reference format error__: `\citep{}` should be used for references w.r.t. the ICLR submission style.

2. __Lacking proper references__: In the Introduction section Lines 80-81, the authors state that previous works on dynamic token pruning favour Gumbel-Softmax since RL-based methods converge slowly. Could the authors cite which existing work(s) claims this? Besides, in the Introduction section Lines 87-88, MAPPO is mentioned as a representative RL-based method, but without reference.

3. __Missing quite a lot of important token reduction works__: In both the Introduction and Related Work sections, only a few fundamental yet outdated token reduction methods are cited. EViT [1] that proposes an efficient token selection strategy based on the [CLS] attention should be mentioned and compared as a strong baseline. ATS [2] that utilizes a learnable scoring function for estimating the importance of each token should be mentioned and compared as a strong baseline as well. In addition to [1,2], many following token pruning methods [3,4,5] and token merging methods [6,7,8] should be included and possibly compared in this paper.

4. __Insufficient experiments__:

    4.1. __Lacking backbones__: The proposed RL4DViT is only adopted and validated on DeiT-B [9]. However, its performance on other backbones is unclear. To demonstrate its __generalizability on different model sizes__, experiments on DeiT-S and/or DeiT-T should be conducted. To demonstrate its __generalizability on different ViT architectures__, experiments on LV-ViT [10] or Swin-Transformer [11] should be conducted.

    4.2. __Lacking runtime comparisons__: Although this paper provides theoretical computational complexities (i.e., GFLOPs), these complexities do not indeed reflect the model's efficiency. Some methods with low GFLOPs may result in an even longer inference time since some operations (e.g., tensor reshaping, and in-memory selection) do not count toward the theoretical complexity [12]. Following the latest common practice, I suggest the authors report the real inference time.

5. __Lacking motivations on using multi-agent RL__: When utilizing MAPPO in RL4DViT, the authors adopt the parameter-sharing schema for agent policies and value functions. Thus, it arouses an intuitive question that whether current MAPPO can be replaced by __single-agent__ PPO. This question is not well addressed from the perspective of both the Introduction and Method parts. In addition, in the experiments, owing to the utilization of MAPPO, both PPO and IPPO [13] should be included in the baseline methods to illustrate the advantages of the multi-agent framework and the centralized critique respectively.

6. __Trivial performance gain__: While DynamicViT [14] achieves 81.3% top-1 accuracy on DeiT-B with 11.2GFLOPs, the proposed MAPPO-DeiT-B only achieves 81.38% with 11.6GFLOPs. Such performance gain is trivial and does not demonstrate the superiority of using RL in token selection. Nonetheless, DynamicViT is a 2021 work and has been surpassed by many following works in both accuracy and efficiency.

7. __Lacking in-depth analysis__: Following Weakness 6, given that this paper focuses on the token selection part, the authors should justify why the MARL-based selection is better, with comparisons to other token selection strategies outlined in [1,2,6,7,8]. However, this paper lacks a quantitative analysis of the benefits of using MARL. And the qualitative analysis in Figure 2 does not clearly demonstrate its advantage over existing token selection methods.

[1] Liang, Youwei, et al. "Not all patches are what you need: Expediting vision transformers via token reorganizations." ICLR, 2022.

[2] Fayyaz, Mohsen, et al. "Adaptive token sampling for efficient vision transformers." ECCV, 2022.

[3] Xu, Yifan, et al. "Evo-vit: Slow-fast token evolution for dynamic vision transformer." AAAI, 2022.

[4] Kong, Zhenglun, et al. "Spvit: Enabling faster vision transformers via latency-aware soft token pruning." ECCV, 2022.

[5] Kong, Zhenglun, et al. "Peeling the onion: Hierarchical reduction of data redundancy for efficient vision transformer training." AAAI, 2023.

[6] Bolya, Daniel, et al. "Token merging: Your vit but faster." ICLR, 2023.

[7] Kim, Minchul, et al. "Token fusion: Bridging the gap between token pruning and token merging." WACV, 2024.

[8] Xu, Xuwei, et al. "GTP-ViT: Efficient Vision Transformers via Graph-based Token Propagation." WACV. 2024.

[9] Touvron, Hugo, et al. "Training data-efficient image transformers & distillation through attention." ICML, 2021.

[10] Jiang, Zi-Hang, et al. "All tokens matter: Token labeling for training better vision transformers." NeurIPS, 2021.

[11] Liu, Ze, et al. "Swin transformer: Hierarchical vision transformer using shifted windows." ICCV, 2021.

[12] Haurum, Joakim Bruslund, et al. "Which tokens to use? investigating token reduction in vision transformers." ICCV, 2023.

[13] Witt, De, et al. "Is independent learning all you need in the starcraft multi-agent challenge?." arXiv preprint arXiv:2011.09533 (2020).

[14] Rao, Yongming, et al. "Dynamicvit: Efficient vision transformers with dynamic token sparsification." NeurIPS, 2021.

**Questions:**

1. According to W2, could the authors please provide references/citations to the claim in Lines 80-81 and which MAPPO method is adopted?

2. According to W3, could the authors please discuss these token reduction methods in the paper and compare the proposed RL4DViT with them? Moreover, could the authors please further compare different token selection strategies since this paper mainly focuses on the token selection part?

3. According to W4, could the author please provide more experimental results on different backbone sizes and different backbone architectures? Could the authors please provide real running time comparisons, especially with EViT, ATS and ToMe?

4. According to W5, could the authors please provide a clear justification for using multi-agent PPO over single-agent PPO? Could the authors please conduct experiments to validate this choice?

---

> ### Author Response · Authors · 2024-11-13
>
> Thank you very much for taking the time to review my paper and for providing valuable feedback. Your suggestions have helped me better understand the shortcomings in my paper, particularly regarding the baseline and the expansion of the backbone. I will carefully revise the paper according to your advice to improve its quality.
>
> I have decided to withdraw my submission to make further revisions. Once again, thank you for your hard work and guidance.

---

### Official Review · Reviewer_82iW · 2024-11-02

**Soundness:** 2
**Presentation:** 1
**Contribution:** 2
**Rating:** 3
**Confidence:** 4

**Summary:**

Vision Transformers have a fixed token sequence that is independent of the input. However, computer vision tasks vary in complexity and the amount of token require maybe dependent on the input. This paper explores dynamic token pruning using an RL approach, and demonstrate that this reduces the computational cost of ViTs. They apply Multi-Agent Proximal Policy Optimization (MAPPO) to determine at each layer of a ViT whether a subset of token should be discarded. They claim to be the first work that integrates RL for dynamic token selection in ViT models. Experiments are on ImageNet, and show that their method reduces computational cost by 39% with a 0.17% decrease in accuracy.

**Strengths:**

- The motivation is clear why we want to prune tokens in large vision transformer models and an RL approach seems like a reasonable solution.
- Experiment results on ImageNet are interesting, showing that their approach does reduce redundant / unnecessary tokens in the image thereby reducing the computational cost without compromising performance.

**Weaknesses:**

- Several issues with spacing and formatting throughout the paper.
- Typos in Figure 1. “specified”, “zeroed-out tokens” and many other typos throughout the manuscript.
- Experiments are only conducted on ImageNet which calls into question the scalability / applicability of the RL4DViT approach on other computer vision tasks and datasets.
- Little motivation provided for the design decisions in formulating the Multi-Agent MDP problem. For example, why is each token an agent in the environment?
- Too many unnecessary RL implementation details provided in the methods section. Equation 1 and 2 are simply the actor and critic losses for PPO training and doesn’t feel necessary.
- What does it mean for an agent to be alive in the reward function definition? Also it is not clear why these agents are both competitive and cooperative? It is not clear why the agents’ actions are not independent of one another.
- Instead of framing this as a MAMDP, could this just be a single agent that predicts the full binary mask?
- Can Figure 2 also show what the token selection is for the baseline methods so there is a qualitative comparison?
- In Table 1, it looks like the base Deit-B model has a higher GLOPs and better Top-1 Acc on compared to the proposed approach.

**Questions:**

- Why did you choose to model this as a multi-agent RL problem as opposed to a single agent?
- Can you explain the design choices for constructing the MDP in more detail?
- Can you provide more experimental results in other computer vision tasks and datasets?

---

> ### Author Response · Authors · 2024-11-13
>
> I want to sincerely thank you for the time and effort you dedicated to reviewing my paper. I greatly appreciate your insightful feedback and suggestions. I will carefully revise the paper in line with your advice to improve its quality.
>
> I have decided to withdraw my submission to make further revisions. Your feedback has been invaluable in helping me recognize areas that need more work, and I am committed to addressing these issues.
>
> Thank you again for your support and guidance throughout this process.

---

### Official Review · Reviewer_fmu9 · 2024-11-03

**Soundness:** 1
**Presentation:** 1
**Contribution:** 2
**Rating:** 1
**Confidence:** 2

**Summary:**

This paper models the token selection acceleration for ViT models as a Multi-Agent Reinforcement Learning problem. Compared with the unaccelerated baseline, the proposed method reduces the computational cost while largely maintaining the performance on the ImageNet classification benchmark.

**Strengths:**

- This paper models the token selection problem in ViT model as a Multi-Agent RL process. Though I am unfamiliar with this field, I think it is a novel attempt.

**Weaknesses:**

- I think the most significant weakness of this paper is its presentation.
  - This paper doesn't give much insight into the advantage of modeling token selection as a MARL process, nor does it explain how the token selection procedure can be transformed into an RL formulation. The paper only lists the computing procedure, which makes the methodology hard to follow.
  - Additionally, there are many repetitive rows in both tables. The paper does not give any explanation of the meanings of these rows. This makes me quite confused.
- I am also confused about the performance of the proposed method.
  - Is the performance of the proposed method better or worse than DynamicVit? The paper only provides two tables, but doesn't present any explanation for the meaning of the figures in the table.

**Questions:**

- Will more insights be provided so that the paper could be easier to follow?
- What's the meaning of the repetitive rows in the tables?
- Is the performance of the proposed method better or worse than
- I also have a concern that the token selection approach is effective mainly because the ImageNet classification problem is too easy. Would it also be effective on more challenging tasks?

---

### Note · Authors · 2024-11-15

I have read and agree with the venue's withdrawal policy on behalf of myself and my co-authors.